

# RF-interference detection and mitigation in the DWD C-Band weather radar network

Maximilian Schaper[1], Michael Frech[1], David Michaelis[2], Cornelius Hald[1], and Benjamin Rohrdantz[2]

[1]Deutscher Wetterdienst, Meteorologisches Observatorium, Albin-Schwaiger-Weg 10, 82383 Hohenpeissenberg, Germany
[2]Deutscher Wetterdienst, Niederlassung Hamburg Sasel, Frahmredder 95, 22393 Hamburg, Germany

**Correspondence:** Maximilian Schaper (maximilian.schaper@dwd.de)

**Abstract.** C-Band weather radar data are commonly compromised by interference from external sources even though weather radars are the primary and therefore privileged user of this frequency band. This is also the case for the radar network of the German Weather Service (Deutscher Wetterdienst, DWD). Theoretically, dynamic frequency stepping (DFS) by devices operating in the C-Band should prevent any disturbance of the primary user. In practice, this does not always work as intended

by the current regulations. As it is not possible to separate a superimposed interference signal from measured weather radar data, the protection of the frequency band is of utmost importance. Currently the only available option is to discard the compromised portions of the radar data. Therefore, the current best course of action is to shut down interference (RFI) sources as fast as possible. The automated RFI detection algorithm for the German C-Band weather radar network is operational since July 2017, which makes use of routinely measured radar moments. Built upon the data gathered since 2017, an RFI classification

with respect to the severity and duration of RFIs was first implemented in 2019. An independent verification of the RFI detection algorithm was performed by using a commercially available WIFI adapter, which is directly integrated into the radar receiver. Subsequently, a mitigation workflow was implemented to efficiently identify and shut down detected RFI sources by the German Federal Network Agency (BNetzA). By following this workflow with great effort, the number of persistent RFIs is decreasing since October 2019 while a steady increase in short lived RFIs over the last 5 years exists. In total, 11889 RFIs

have been identified since July 2017 until May 2022. The majority of these (94.8 %) are so short lived that an unambiguous identification by the BNetzA is, in general, not feasible. However, as stated by the C-Band regulations, any non-compliant transmitter compromising the operation of a weather radar has to be shut down. This is important, as even these short lived RFIs negatively affect the meteorological product generation.

## 1 Introduction

Data obtained by weather radars are a crucial element in today's forecast and nowcasting tools (e.g. Besic et al., 2016; Steinert et al., 2021). Nowcasting applications and the analysis of local phenomena depend upon the high temporal and spatial resolution provided by weather radars. Forecast models assimilate radar data. The spatial distribution of the precipitation amount, which is an essential element for a wide range of hydrological services, is only available if radar data are available. As such, the 17 operational C-Band weather radars are a crucial part of the observation network operated by the German Meteorological



Service (Deutscher Wetterdienst, DWD; Frech et al., 2013). Data gathered by weather radars have to be available at all times and satisfy strict quality criteria (Frech, 2013; Frech et al., 2017).

Weather radars are the primary and therefore privileged user of the $5600 - 5650$ MHz frequency range of the C-Band in Germany and other countries (ETSI, 2012). A primary user must not be disturbed by other devices operating in this frequency band. The World Radiocommunication Conference in 2003 (WRC-03) allowed WIFI in the C-Band, if a dynamic frequency
stepping (DFS) is implemented (ITU-R, 2003). In this paper, so called *WIFI networks* include radio local area networks (RLAN), wireless local area networks (WLAN), as well as wireless access systems (WAS) especially in the 5 GHz range. It was believed that a coexistence between weather radars and WIFI devices would be possible. However, if a user checks e.g. a current European weather radar composite, most of the time there will be regions with typical line-shaped reflectivity signals, which most commonly originate from WIFI networks operating in the C-Band (Huuskonen et al., 2014). Weather radars are
increasingly compromised by these active radio frequency interference (RFI) sources, which clearly indicates that the intended coexistence in the C-Band is not always working (Saltikoff et al., 2016; Palmer et al., 2021). The reason for this are often non-compliant devices on the market which do not have a properly working DFS, while in some cases the DFS is disabled entirely (ECC, 2014). It is the task of the national regulation agencies (NRA) to enforce the proper coexistence. The German Federal Network Agency (Bundesnetzagentur, BNetzA) starts to act upon an RFI being actively reported. A field team is dispatched
by the NRA to identify the RFI source, which in our case is done in close collaboration with the DWD radar team. The RFI source will be shut down by the NRA as soon as it has been identified by their field team, mitigating any further interference caused to the radar. A more detailed aggregation of the regulatory history and interference situation all over the world can be found in Saltikoff et al. (2016).

The efficient and objective detection and removal of external radio frequency (RF) transmitters that disturb radar data is
45 an ongoing and time consuming challenge (see also Saltikoff et al., 2016; Huuskonen et al., 2014; Carroll et al., 2010). The removal of an RFI source is essential, because the raw meteorological information gets masked by an external interference and can not be properly recovered. This is especially true for quantitative products derived by fully automatic algorithms. Up to date there is no algorithm available which can separate the meteorological signal from a superimposed RFI signal in the raw radar data (IQ data) on the radar signal processor. On the other hand, algorithms to identify RFI signatures in the derived radar
moments do exists and are in most cases somewhat able to increase data quality by applying strict filter methods (e.g. Cho, 2017; Peura, 2002).

This paper introduces the RFI detection algorithm developed and used by DWD to prioritise and mitigate as many RFIs as possible. The algorithm utilises thresholds to determine if a specific ray is contaminated by an RFI and groups such contaminated rays in order to recognize RFI sources. The presented RFI detection algorithm is then verified by using a so called
"RHunt", which is based on a commercially available WIFI adapter (see Chapter 3). To make the verification as similar as possible to the operational weather radar measurement, the actual radar antenna is connected to the WIFI adapter during measurements. By using the independent WIFI adapter, this method provides a reference data set of potential RFI sources that operate at C-Band within view of the radar. The resulting RHunt data is then compared to automatically detected RFI sources based on actual weather radar data.





Chapter 2 describes the RFI detection algorithm which is used to identify and categorise RFI sources in the radar data to efficiently support the RFI mitigation process. These identified RFI sources are reported to the German NRA (BNetzA). In collaboration with DWD, a team of NRA technicians use the provided information to efficiently identify and shut down the RFI sources. This is still a time consuming process, since an RFI source has to be identified unambiguously by measurements of the NRA field team before it can be shut down.

In addition to the RFI detection algorithm, we also provide an analysis into the number, frequency and severity of detected RFI sources encountered since July 2017 when the monitoring of RFIs was operationally implemented for the German radar network (see Chapter 4). In the conclusion we summarize the main outcome and provide an outlook on future steps.

## 2 Identification of an RFI

Reliably identifying RFIs within the radar data is the first step to getting rid of this impairment of data quality. Until 2019, the
70 task of RFI identification and reporting was done by radar data users and customers. If an RFI had caused enough problems to a meteorological product it had to be reported manually. This required the users to identify the source of the problem as external interference and qualify the resulting impairment of the specific product subjectively by eye. As data quality of raw measurements gained more weight, an automated and objective procedure was required. Therefore, the now operationally used RFI detection algorithm was developed, while also keeping the established procedure for users and customers to independently
report RFIs.

  To identify interference within the radar data, we use the signal quality index (SQI) and normalized standard deviation of the received power (STD). This follows the approach used in the weather radar signal processor ENIGMA developed by GAMIC to identify interference in radar data (latest model GAMIC, 2022). SQI quantifies the coherence of the received signal with the transmitted pulse and thereby indicates if the received signal originated from the transmitted radar pulse or from an external
transmitter. STD is the normalized standard deviation of received power in a batch of pulses which are aggregated into the same ray. Typically, about 50 pulses are aggregated within one ray (ray azimuthal width is 1°). STD is evaluated in real-time by the GAMIC signal processor for each range bin. Using a similar kind of measure to detect interference has also been suggested by other studies (e.g. Rojas et al., 2012; Keränen et al., 2013; Cho, 2017). A large STD is indicative of a fast varying signal power, which can be related to pulsed communication signals from an external source. STD is low if there are only small continuous
variations in signal power between received radar pulses.

  A large STD and a small SQI in combination are indicative of an external WIFI interference. When an external WIFI is present, either a specific ray or a full sector of multiple rays may be disturbed by a single RFI source (see also Saltikoff et al., 2016). This depends on how the radar is picking up the WIFI signal, which can either happen through the radar main lobe or in case of strong interference even through sidelobes (see also Section 3.2). In consequence, close or high powered RFI sources
most likely contaminate an entire sector and not only a single ray.

  Figure 1 show an example of compromised operational weather radar data including precipitation. It contains four PPIs from the same 0.5° elevation sweep. The sweep was taken at the DWD radar site Isen (ISN) at 15 March 2022. One can clearly see





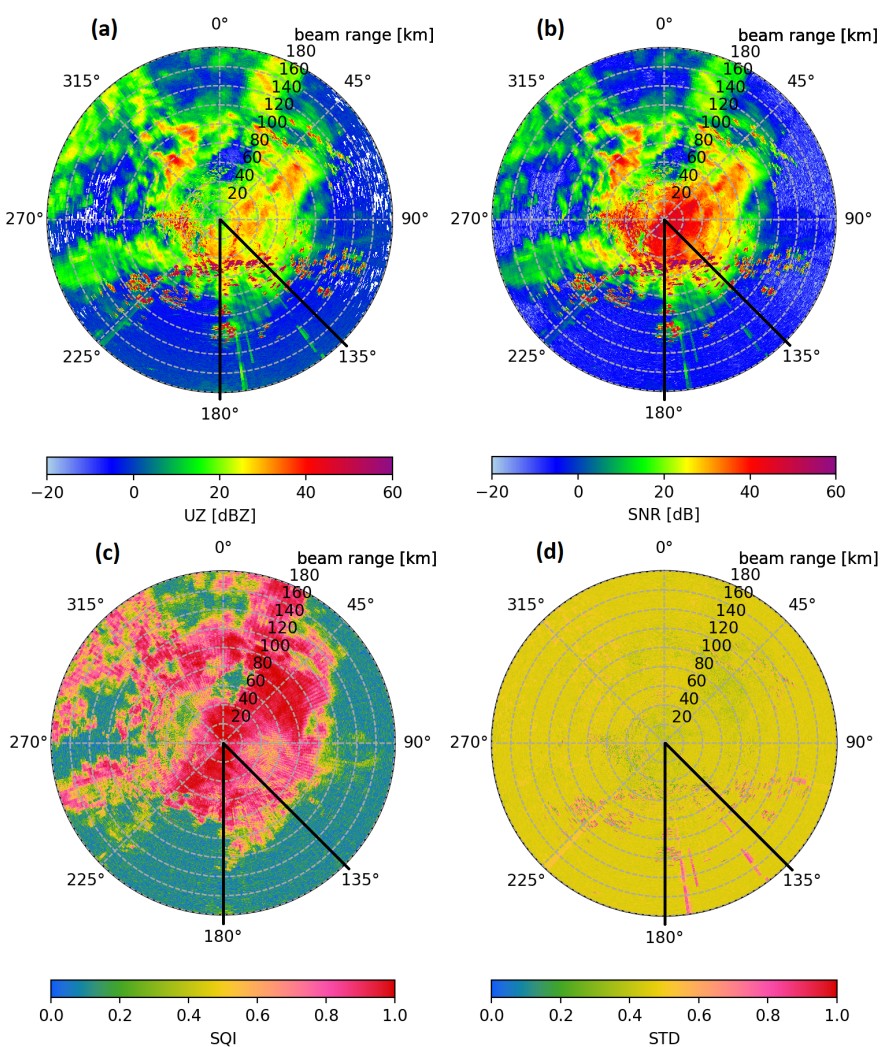

**Figure 1.** Example PPIs from radar site Isen of the 0.5° volume sweep at 15 March 2022, 15:32. Shown are uncorrected reflectivity (UZ), signal to noise ratio (SNR), signal quality index (SQI) and pulse based normalized standard deviation (STD). Within the azimuth section from 135° to 180° (marked by the black lines) multiple linear signatures can be seen in a, b and d, caused by external interference. The effect of another weaker inference can be seen near 225°.

the meteorological signals with an uncorrected reflectivity (UZ) of over 10 dB in (a), an SQI greater 0.6 in (c) and a low STD of 0.3 − 0.5 in (d). Dark red spots in UZ and pink speckle in STD south of the radar site are caused by a mountain range as ISN is located just north of the Alps. The sector in the southeast marked with black lines contains several RFI signatures. These RFI signatures are most visible in the STD (d) as linear structures. Looking at UZ (a) as well as the SNR (b) one can see that the signal strength for both the RFI and meteorological signal are on the same order of magnitude in the outer regions. Meanwhile,





close to the radar site, the up to 40 dB strong meteorological signal dominates over the external interference signal. One can also identify a localised decrease in SQI due to the RFIs for a few rays, which overlaps with the effect of the staggering.

To objectively identify if a ray is "disturbed" (containing interference from an external source) threshold values were chosen to characterise each ray. Using the above mentioned radar moments, a ray is identified as disturbed if mean SQI is below 0.6 and mean STD is greater 0.6 over all range bins of the individual ray. These disturbed rays are collected and grouped to identify RFI sources which are then reported to the NRA. The following Section 2.1 outlines the RFI detection algorithm, while further technical details of the algorithm can be found in the appendix Section A. Section 2.2 describes the information resulting from

the RFI detection algorithm.

### 2.1   Processing disturbed rays to detected RFIs

The first step to detect RFI sources is the identification of "disturbed" rays. As mentioned before, we identify a ray as "disturbed" (containing interference from an external source) if mean SQI is below 0.6 and mean STD is greater 0.6 over all range bins of the individual ray, which builds upon the interference detection of the GAMIC signal processor (latest model GAMIC,

2022). The identification of disturbed rays is carried out for the precipitation scan (DWD's terrain following scan, dedicated to precipitation estimation) and the lowest three elevations of the volume scan (0.5°, 1.5° and 2.5°). Every 5 minutes, when one full scan cycle finishes, this identification is executed at each radar site. Hereby, all of the following parameters are stored for each ray satisfying the described criteria: timestamp, azimuth, elevation, SQI, STD, signal to noise ratio (SNR).

    An aggregation of disturbed rays is then transferred to a central processing site, where they are evaluated once a day for

all radar sites of the DWD radar network. Each site is evaluated individually. The main focus of the mitigation process are persistent RFIs which potentially have the strongest impact on data quality and also have the highest chance to be successfully identified and removed by the NRA. To process the data, the following five steps are carried out (further technical details can be found in appendix Section A):

1. **preparing the data** - All disturbed rays within the last 30 days are collected.

2. **combining disturbed rays** - Collected rays are grouped into small "RFI ray boxes" by proximity in time and azimuth for each radar site. Such an "**RFI box**" is defined by an upper and lower boundary in azimuth as well as a specific start and end time, resulting from the outermost disturbed rays captured in the box. At this stage only disturbed rays in close proximity are grouped together, resulting in these initial RFI ray boxes.

3. **joining RFI boxes** - The small RFI ray boxes are further joined in azimuth and time using an iterative process.

4. **characterising final RFI boxes** - Final RFI boxes are classified and categorised (described in detail below).

5. **result visualisation and aggregation** - The final RFI boxes each contain detailed information relating to a single RFI source. A scatter plot for the entire 30 day period is created for each radar site. It contains a dot for each disturbed ray as well as *relevant* final RFI boxes in red (see Figure 3). In addition, an HTML-table is generated containing detailed



**Table 1.** Listed are three classifications which are used to describe an RFI. The classification is done by evaluating the most disturbed azimuth angle from the RFI box and determining if the fraction of these specific disturbed rays exceeds the specified threshold. An RFI may fall under multiple of these RFI classes. The sum of reference values uniquely identifies the combination of applicable RFI classes.

| RFI class | reference value | select rays with mean SNR | fraction of disturbed rays | impact on "RFI severity" |
|---|---|---|---|---|
| strong | 4 | $> 20\,\text{dB}$ | $\geq 1\,\%$ | $+5$ |
| persistent | 2 | $> 0\,\text{dB}$ | $\geq 10\,\%$ | $+5$ |
| weak | -1 | $< 5\,\text{dB}$ | $\geq 10\,\%$ | $-2.5$ |

information about all *relevant* RFI sources for all sites. The table includes the characteristics of *relevant* RFI sources as
well as the current processing status regarding their removal.

The final RFI boxes each describe a single RFI source. To allow for a simple interpretation of the final RFI boxes they are classified and categorised regarding their relative threat level to data quality. First a classification of each RFI box is performed. The resulting "**RFI class**" characterises the rays contained within the RFI box. Table 1 lists the three possible categories and their numeric reference values. It can be any combination of: strong, persistent and weak. Each RFI box is classified by
135 evaluating the data from the most repeatedly disturbed (worst) azimuth direction contained within the RFI box. For example, "strong RFI" means that at least 1 % of the disturbed rays within the worst azimuth direction have a mean SNR over 20 dB. It is possible that an RFI is classified as strong and weak at the same time, which most likely indicates a varying signal power level emitted by the RFI source.

To further characterise and prioritise the detected RFI sources, another measure was developed: the "**RFI severity**". The
140 RFI severity is an empirical measure to automatically and objectively assign a priority to a detected RFI source for tracking and reporting. The relationship between the internal RFI box parameter and the resulting effect on the RFI severity is denoted as "severity impact". Several internal parameters of a detected RFI are factored in to calculate the RFI severity. Each of these factors is evaluated regarding the specific severity impact (see Figure 2). If an internal parameter exceeds the described range, the respective maximal or minimal severity impact is used. All of these factors are added up to arrive at the final RFI severity.
The following is a description for each of these severity impact factors:

**strong RFI** *— RFI class —* used if there are a few rays with high SNR.

**persistent RFI** *— RFI class —* used if a considerable amount of rays are affected daily.

**weak RFI** *— RFI class —* used to indicate a considerable amount of rays with low SNR.

**"worst" SNR** *— mean SNR of rays in worst direction —* The measured meteorological signal gets superimposed with the
150 interference signal. Separating these signals back into there components is not possible. Therefore, the strength of the interference signal directly impacts the ability to recognise the valid meteorological signal in the presence of the inter-





ference. The keyword "worst" describes which rays are used to compute this parameter: rays from the most repeatedly disturbed azimuth direction contained within the RFI box.

**"worst" disturbance** — *mean disturbance fraction in worst direction* — To distinguish sporadic and persistent RFIs, two severity impact regimes are defined for this parameter (see Figure 2). These at most sporadic RFIs get a high severity penalty, as resolving the cause of such RFIs with the NRA is unlikely. In addition, it is most likely that such sporadic RFIs will disappear without any intervention.

**mean disturbance** — *mean disturbance fraction over all azimuth directions* — To interpret this factor one has to consider the azimuthal width of RFI boxes. For example, a 1° wide interference with 100 % disturbance, which in a single time step also affects a neighboring ray, will result in a final RFI box with an azimuthal width of 2°. Therefore, the mean disturbance would be calculated to about 50 %. As this effect is common for most RFIs, the mean disturbance is rather low in general.

**number of rays** — *total number of disturbed rays* — The exact number of disturbed rays is more relevant for short term RFIs containing a few hundred rays, than it is for long term RFIs containing several thousand rays. Therefore, this factor is only used to differentiate small RFI boxes from one another with a maximum of 576 rays (2 days worth of rays in a single direction), after which the maximum severity impact of 10 is applied.

**azimuthal width** — *width in azimuthal direction* — This factor, regarding only the azimuthal width of an RFI, contributes little to the overall severity, as to not overrate wide, sporadic RFIs. Wide RFIs appear rarely and most likely in conjunction with a high mean disturbance if they pose a high threat to data quality.

**duration** — *duration for which the interference was active in days* — The duration is one of the main characteristics in deciding whether or not the RFI is reported to the NRA. Most RFIs are short lived and disappear by themselves. Reporting these to the NRA as RFIs to be traced is inefficient in the sense that the RFI source has a low chance of being found.

**last activity** — *time in days since the last disturbed ray was added* — Similar to the duration, this factor helps in operational decision making to filter out RFIs which already ceased to exist. This factor is excluded from the historical analysis, as it is irrelevant how long ago any specific RFI ceased to exist.

To given an example, one of these factors is the daily *mean disturbance* fraction, describing the fraction of disturbed to undisturbed rays. In our case (with a scan cycle repeating every 5 minutes), there will be 288 rays with the same azimuth and elevation during one day. Therefore, the factor is computed by mean number of daily disturbed rays with respect to 288 rays. This also means, if the radar was out of service, a *mean disturbance* of 100 % can not be reached, as the service time is not accounted for and no disturbed rays could be measured during a maintenance period.

The severity impact of these factors is chosen to reflect the relative impact on data quality caused by the different aspects of individual RFIs. In addition, the chosen severity impacts account for underlying differences resulting from short term versus




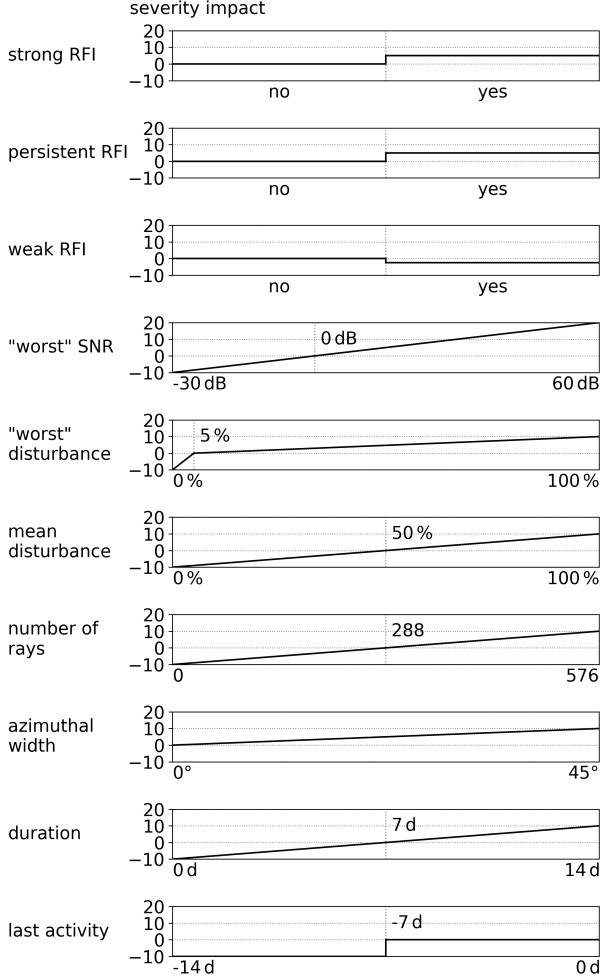

**Figure 2.** A panel for each of the factors used to determine the overall *RFI severity* is shown. Each panel relates the underlying variable to a resulting *severity impact*. The top three panels relate to RFI classifications. All other panels relate to specific parameters of an individual RFI and contain min. and max. boundaries. Outside of the specified boundaries the last valid severity impact is used.

long term interference. Another point to note is that some aspects of an RFI are reflected in multiple of these severity factors (e.g. the amount of caused disturbance).

By using the RFI severity three "**RFI categories**" were defined: moderate, severe and critical. RFIs with a severity above zero are considered to be *moderate* and have a noticeable impact on data quality. Any RFI with less than zero severity is not actively tracked by the DWD as there is, in general, no hope of a successful identification and removal by the NRA. Further details regarding this fact can be found in Chapter 4. RFIs with a severity greater 10 are considered to be *severe*, while an RFI with a severity greater 25 is considered to be *critical*. Critical RFIs are to be mitigated as soon as possible by direct



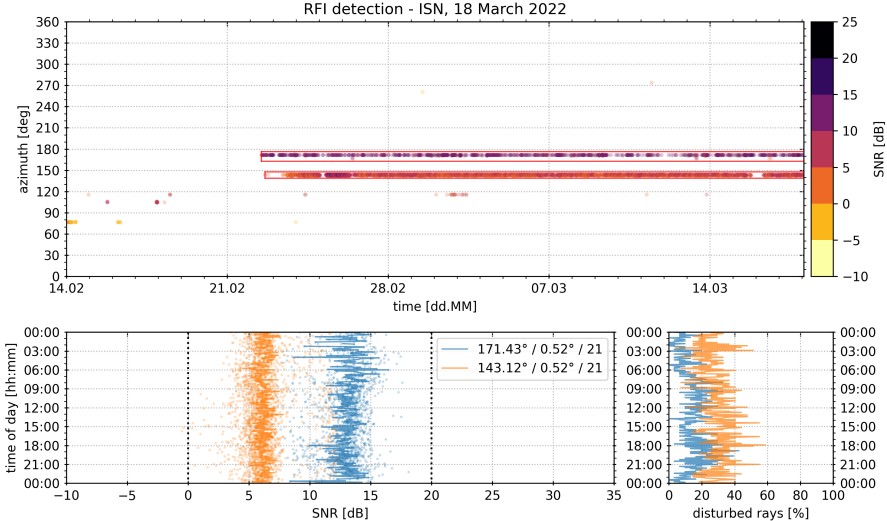

**Figure 3.** The operational RFI analysis from 18 March 2022. Included are disturbed rays from the previous 30 days. The top panel shows each disturbed ray as a dot in the scatter plot (color of dots depicts the median SNR). All detected RFIs with an RFI severity greater zero are marked with a red box. At the bottom additional properties of each severe RFI are displayed. The data in both bottom panels is accumulated into 5 min bins according to the time of day (y-axis). The bottom left panel shows SNR and the bottom right the fraction of disturbed rays. Within the legend the mean azimuth, mean elevation and RFI severity of each severe RFI are displayed.

communication with the NRA, while moderate RFIs are tracked with a low priority in respect to other tasks of the DWD radar team.

## 2.2  Data resulting from the RFI detection

After collecting all rays with an interference signal and joining them into RFI boxes, an automated visualization and summary of the results had to be developed. These aggregated results are used by the technical DWD personnel and the NRA. Therefore, the main requirement was to provide an intuitive representation of the detected RFIs to support quick decision making and the subsequent removal process of the RFI source in the field. An example of the final visualisation from the radar Isen (ISN) is shown in Fig. 3. All of the relevant information for the radar technicians to support the mitigation process is summarized graphically. In the top panel of Fig. 3, each disturbed ray is plotted as a dot in time and at the respective azimuth. Each dot is color coded indicating the median SNR of a single disturbed ray. All *relevant* RFI sources (RFI boxes) above the zero RFI severity threshold are marked as red boxes. The lower two panels provide further quantitative information on the temporal variability and strength of the *relevant* RFI sources. Both bottom panels display the data in relation to the time of day on the y-axis. This has been very valuable in setting up an onsite appointment with the NRA at a specific time of day to ensure that the RFI source is actually active and can be found by the NRA.





In this case, there are two persistent RFIs present. One at a mean azimuth of 171.43° (blue in lower panels) with a SNR in
the order of 7 dB which is present about 25 % of the time and another RFI at 143.12° with a SNR in the order of 13 dB which
is only present about 15 % of the time. There is no significant diurnal variation visible for any of the two RFIs. Therefore,
no specific time schedule is needed when the NRA is onsite to identify the sources of interference. Both RFIs are assigned a
severity of exactly 21, which makes both of them severe RFIs. This also shows how the differences in strength (bottom left)
and persistence (bottom right) of both RFIs balance one another regarding the resulting RFI severity.

Further explanations and details to get a better understanding of the RFI detection results can be found in Chapter 4. There,
a statistical analysis over the nearly 5 years of DWD RFI data is performed, which offers a deeper understanding of the RFI
severity, the chosen severity limits to categorise detected RFIs and long term trends.

## 3 Validating the RFI detection algorithm

For an independent verification of the RFI detection algorithm, a commercially available WIFI adapter is integrated into the
radar system. It was first built into DWD's research radar at the Hohenpeißenberg Observatory (MHP), where the first dedicated
verification experiment was carried out. In addition, similar verification measurements have been carried out at the operational
radar Isen (ISN), where the RFI detection algorithm identified two active RFIs (see Figure 3). The objective was to gather a
full sweep of verification data at 0° elevation, record all visible WIFI networks and identify those which operate at C-Band, as
those are the most likely cause of radar interference. The gathered verification data are then compared to the results from the
RFI detection algorithm and raw radar data. Results of two case studies are shown in Section 3.2 and Section 3.3. The practical
experience from over two years of operation, which includes the removal of RFIs by the NRA based on this methodology,
indicates a good reliability of the RFI detection algorithm and classification approach.

### 3.1 RHunt setup and WIFI regulations

The hardware setup to collect an independent data set is shown in Fig. 4. A commercially available WIFI receiver from NETIS
(WF2190 AC1200) is currently integrated into the radar receiver and can be turned on remotely. Instead of using the two
supplied antennas (the WIFI adapter has to antenna ports), each input port is connected to a directional coupler behind the low
noise amplifier (LNA) of the radar receiver and is therefore connected with the radar antenna. This has the advantage that the
same analog channel is in use for both the verification measurement and normal radar operation. The commercial WIFI adapter
is plugged into a Raspberry Pi 3, which is used to control and acquire the data stream. We call the combination of commercial
WIFI adapter and Raspberry Pi the "RHunt" (short for RLAN Hunter). The RHunt permanently remains in the radar receiver
chain and is operated remotely. During normal radar operation the RHunt power supply (Acromag) is turned off to minimize
potential self inflicted interference from the integrated WIFI transmitter. Details regarding the actual data acquisition process
can be found in the appendix Section B.

The positioning of the couplers connected to the WIFI adapter in the radar receiver chain is critical. To minimize a negative
impact on the SNR during normal radar operation, additional hardware components should not be installed before the LNA.



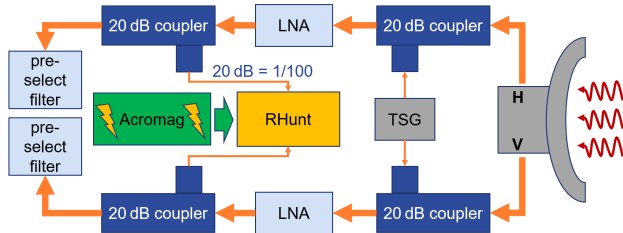

**Figure 4.** Diagram of the RHunt setup within the DWD radar system. The RHunt is integrated into the analog receive path of the weather radar after the low-noise-amplifier (LNA) and connected in horizontal (H) and vertical (V) through a 20 dB coupler in front of the pre-select filter. The Acromag serves as a remote controlled power toggle.

Additionally, to obtain comparable results between radar and WIFI receiver the coupler should be placed as far down the receiver chain as possible, meaning right before the down converter (mixer). However, any filters integrated in the radar receiver chain may corrupt the WIFI signal by cutting part of the used band away and thus render it impossible for the WIFI adapter to identify any networks. In case of the DWSR 5001 radar, a 22 MHz pre-select filter is in use, which already has a strong impact
on WIFI networks using up to 80 MHz of bandwidth. Hence, our coupler to the WIFI adapter is integrated right before the pre-select filter but behind the LNA. Thereby, keeping a wide range of signals in the 5 GHz range available to the RHunt which have are picked up by the radar antenna.

The bandwidth of the utilised commercial WIFI adapter covers 2.4 – 2.4835 GHz and 5.18 – 5.825 GHz. In Germany, the commonly used weather radar frequencies 5.6 – 5.65 GHz lie within the C-Band (4 – 8 GHz). Since weather radars and WIFI
may share a common frequency band, a standardized channel allocation scheme was introduced for WIFI access points ETSI (2012). The mandatory mechanism for WIFI devices to prevent interference to weather radars is called Dynamic Frequency Selection (DFS). Since the shared bands vary by country and jurisdiction, an exact geolocation is critical for the DFS to work properly. Upon startup, a WIFI access point will first limit operation to "safe bands" and passively scan shared bands for radar signatures for a predefined time. The required period depends on the country and the actual radar scan intervals in use. If no
radar is detected during this period, the shared bands can be used for WIFI operations. These scans are continuously repeated during the WIFI operation to ensure the continued use of radar free frequency bands. Any frequency changes are communicated by the access point to all connected clients. Generally, the DFS mechanisms works well. However, in some cases DFS does not work as intended and interference arises (see also Carroll et al., 2010; ECC, 2014):

– DFS has been disabled.

– The geolocation can be manually changed, must be manually set by the user, or is even automatically set wrong.

– The WIFI access point is mobile and thus not able to properly detect the radar.

– The WIFI access point is shielded from the radar, but the client is not.

– Sensitivity of the DFS mechanism to weather radar signals is insufficient.





– The radar can have a blanking sector (not transmitting) in the direction of the WIFI access point (see Section 3.2).

Due to the fact that the RHunt is connected into the radar receiver instead of using the commercially supplied antennas, it is not possible to measure an absolute power value for the signal strength of the identified WIFI networks. As the used WIFI adapter already reports the signal strength of each identified WIFI network in an arbitrary range of 0 to 100, that scale is used without further investigation.

However, the most crucial constraint in the RHunt setup is the fact that a WIFI network is only reported by the RHunt if
it is identifiable. This means the WIFI adapter has to be able to recognise the WIFI network as such based on the received signal. Considering that a WIFI source can interfere with the radar measurement without being identifiable, the approach taken is to record all WIFI networks identified by the commercial WIFI adapter. Thereby, including many WIFI networks which are rather far away from the center frequency of the radar. These forced constraints have to be kept in mind during the following analysis.

**3.2    RHunt results from Hohenpeißenberg (MHP)**

The verification data collected with the RHunt contains identified WIFI networks that operate in the 5 GHz C-Band at 5150 – 5725 MHz. For the interpretation of the RHunt results it is important to recognize that this band is split into three distinct frequency bands, which are defined by ETSI (2012). All frequency bands are split into 20 MHz channels which can be combined in order increase the bandwidth. The 5150 – 5250 MHz band is assigned to devices for indoor use which are limited in EIRP
(transmitted power multiplied by antenna gain) to 200 mW. Similarly, the 5250 – 5350 MHz band is also assigned to devices for indoor use with the same maximum EIRP, the only difference being that DFS is required. For those two bands, the frequency block numbers are between 36 and 64. The third band (5470 – 5725 MHz) is considered for indoor and outdoor use. Maximum WIFI EIRP is 1 W and DFS functionality is required. Obviously, WIFI transmitters in this band have the highest potential to interfere with weather radar measurements. Related frequency channel numbers are between 100 and 140.

The Hohenpeißenberg (MHP) radar is operating at a center frequency of 5640 MHz. It is located at the top of a mountain which is about 200 m above the surrounding area. The next bigger cities are in the east and west of the radar site. A radio tower is located on the same mountain range in a direction of about 110° from the radar, which requires a blanking sector (no radar transmission) of about 10° width in azimuth. In addition, two open WIFI access points are present within the blanking sector 250 m east of the radar site as it is a common tourist attraction.

The RHunt results from 22 September 2021 are shown in Fig. 5b. Every identified WIFI is shown as a single dot at the respective azimuth position. The radial position of the dot indicates the signal strength on a scale of 0 to 100. This adds more context to the potential threat level of each WIFI. In addition, all WIFI records above 5500 MHz (channel 100, allowed outdoor) are color coded, while the rest of the WIFI records are displayed in gray scale depending on their frequency. Lines connect the dots with the same MAC-address and therefore originate from the same WIFI source. This particular sweep shows a total of
784 recorded WIFI identifications which showed 127 unique MAC-addresses.



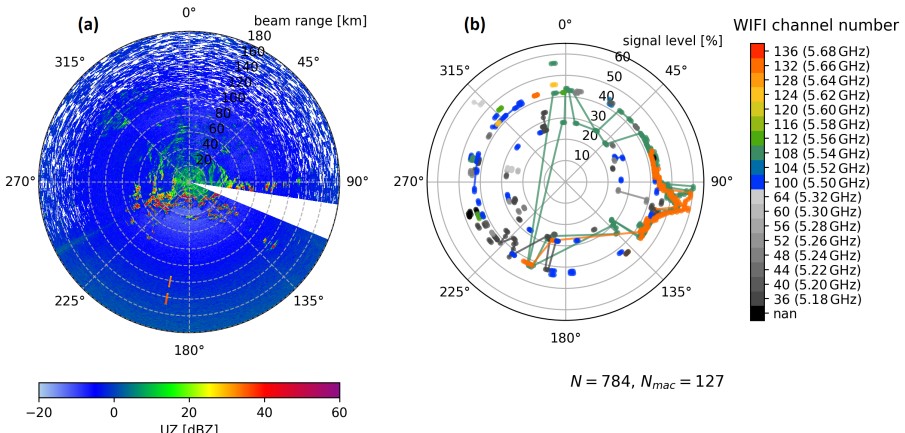

**Figure 5.** An uncorrected reflectivity (UZ) PPI is shown in (a) from the radar site Hohenpeißenberg (MHP). It was taken after the RHunt sweep at 22 September 2021, 18:22. (b) contains the identified WIFI networks at their respective azimuth and signal level as dots. The WIFI channel is color coded. Channels above 100 (5500 MHz, allowed outdoor) are colored, others are in gray scale. Each line connects networks with the same MAC address. In total 784 WIFI records ($N$) and 127 networks with unique MAC addresses ($N_{mac}$) were captured. Note the different variables on the radial axis.

The WIFI records in the azimuth range 90° to 115° stick out with an elevated signal level. Four of these (two turquoise and two orange) were detected over a large azimuth range at varying signal levels, evident by the connecting lines. Two of those networks belong to the mentioned public WIFI access points located about 250 m away from the radar. As noted before, due to the sector blanking the DFS of the WIFI access points in this direction can not see the radar and therefore does not switch away from the radar frequency band. As these WIFI access points are close to the radar site the networks are even identifiable through the sidelobes of the radar antenna.

Figure 5a shows the UZ PPI measured after the dedicated RHunt sweep. Most prominent is the sector blanking in which all of the collected data is discarded. One can also see an interference signature at about 190° azimuth. This signature coincides with RHunt detections of the previously mentioned WIFI networks with elevated signal levels and relates to the known WIFI access points at about 110° azimuth. These WIFI networks must have been picked up by a sidelobe of the radar antenna and are the most likely cause of the visible interference. This showcases a prominent finding: strong or close RFI sources are also visible through sidelobes, which makes it hard to identify the true direction, and therefore the source, of some interference. However, we only see sporadic and rather infrequent interference in the operational MHP radar data mostly at an azimuth of 113°, right at the edge of the sector blanking, where the signal level of the detected WIFI networks is still elevated.

### 3.3 RHunt results from Isen (ISN)

The radar site Isen (ISN) is located in south-eastern Germany and operates at a center frequency of 5625 MHz. In the operational radar data from ISN, two persistent RFIs were detected by the RFI detection algorithm. A RHunt measurement campaign was




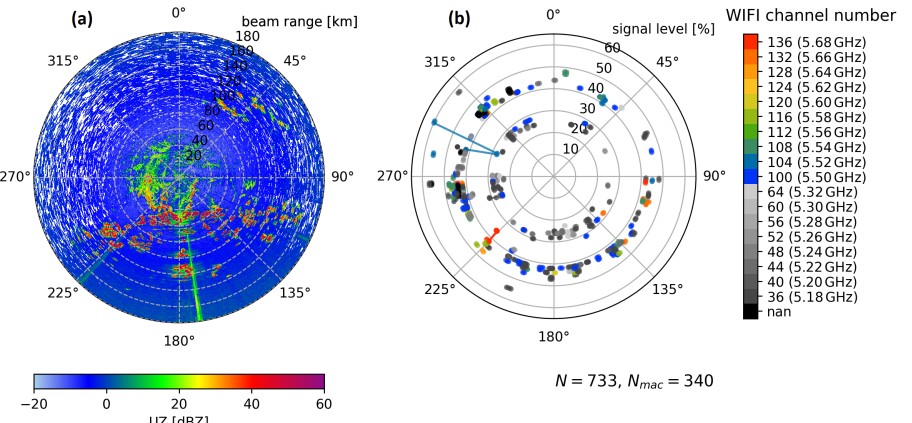

**Figure 6.** An uncorrected reflectivity (UZ) PPI is shown in (a) from the radar site Isen (ISN). It was taken after the RHunt sweep at 6 April 2022, 10:27. (b) contains the identified WIFI networks at their respective azimuth and signal level as dots. The WIFI channel is color coded. Channels above 100 (5500 MHz, allowed outdoor) are colored, others are in gray scale. Each line connects networks with the same MAC address. In total 733 WIFI records ($N$) and 340 networks with unique MAC addresses ($N_{mac}$) were captured. Note the different variables on the radial axis.

carried out on 6 April 2022 in order to make an assessment of the situation, and to provide further information to the NRA. All detected WIFI networks and a corresponding PPI are shown in Fig. 6 using the same layout as in Fig. 5. In ISN, a total of
310 733 WIFI networks were identified which originated from 340 distinct WIFI devices based on the MAC-addresses. This is a considerably larger number of WIFI networks than were found at the MHP radar site (127). The main reason for this may be attributed to the altitude at the MHP mountain site compared to the flat surroundings of ISN with its close proximity to Munich in the west (3rd biggest city in Germany).

The two severe RFIs were detected by the RFI detection algorithm at an azimuth of 143° and 171° (see Figure 3). Only
315 the RFI at 171° is visible in the PPI taken after the RHunt sweep as a linear signature (Figure 6a). The average SNR of this RFI is about 13 dB. What makes it hard to remove is its sporadic nature. Only about 15 % of the sweeps at low elevations are affected, with a small peak in the evening hours. The sporadic nature causes less harm in terms of data quality, but on the other hand it impedes a quick mitigation by the NRA. Focusing on the azimuth range of this sporadic RFI, we find 4 WIFI networks which have a channel number greater than 100 and therefore might be a likely cause for the interference. In contrast to the
320 relevant networks at MHP, the signal level of these four is relatively low, suggesting that these are devices which are not in close proximity to the radar site. Also, none of the four networks are visible over a greater azimuth range.

Even though there is no linear interference signature in the UZ PPI towards 143°, there are multiple WIFI networks recorded (Figure 6). Here one has to know that this particular RFI already ceased to exist on 21 March while the RHunt measurement was executed on 6 April. The RFI was reported to the NRA, but processing stopped shortly after as it ceased without any active
intervention and before an onsite visit. However, this RFI can be seen in Fig. 1 which shows data from 15 March. This shows



how rapidly a severe RFI may arise and disappear without any chance of a definitive identification of the interference source. Although the RFI was present for 26 days and reported to the NRA, the mitigation process for this RFI, with an RFI severity of only 11, was ultimately to slow for an identification of the interference cause.

### 3.4 Discussion of verification

The presented validation approach provides an independent assessment of the presence of WIFI networks using a plug and play approach with minimal limitations to the operational scanning. Operational scanning has to be paused for about one hour to gather a full sweep of verification data. By using the RHunt setup described in Section 3.1, there is no negative effect on radar data quality when the RHunt is powered off. However, the chosen approach has several shortcomings in regards to the collected verification data.

The main drawback of recording WIFI networks from the RHunt setup is that it is only possible to detect networks with an identifiable signature (i.e. MAC-address). Any signal which can not be properly interpreted by the used WIFI adapter goes unnoticed. In addition, the reported properties of the WIFI network are gathered from the received signal and do not necessarily reflect the precise i.e. frequency band in use by the WIFI. Also, in the odd case, any electric installation may cause radio wave emissions visible to the sensitive weather radar. This was the case with an airport lighting installation interfering with the 340 DWD radar site Hannover (HNR). Only WIFI transmitter not complying with DFS requirements or which have out-of-band emissions reaching into the C-Band will be found via the employed method.

Yet another limitation is the reported signal level. One would expect the signal level reported by the commercial WIFI receiver to continuously vary between 0 % to 100 %. As Figure 5 and Figure 6 show, the actual signal level has two dominating levels at around 27 % and 42 %. Here, one has to note that this kind of WIFI adapter is not meant for a detailed analysis of the 345 received signal power, but rather to relay the general information in a consumer friendly manner.

To solve these critical limitations of the RHunt setup, a WIFI spectrum analyzer could be used. These are also commercially available as plug and play systems and we are currently preparing to test three different devices. Utilizing such a WIFI spectrum analyser allows to detect any external signal (including signal power and frequency), without the need to identify the transmitter as a common WIFI in one of the expected frequency bands. However, our goal with this verification was to employ the most 350 common hardware to capture the most likely interference sources. Although a detailed analysis of the spectrum could provide further information, the mitigation time might not decrease as the actual mitigation process currently requires the NRA to confirm the provided information by an onsite visit of NRA field technicians.

Even though the RHunt measurements have proven to be very helpful for the mitigation procedure with the NRA, this verification is ultimately inconclusive. If an RFI detection is followed up by the DWD radar technicians with RHunt measurements 355 one can nearly always see multiple WIFI networks. However, the collected RHunt measurements do not allow to draw a definitive connection between identified WIFI networks and caused interference, which would relate them to the detected RFI. This fact is highlighted by the huge number of detected WIFI networks on frequency bands close to the center frequency of the radar which do not interfere with the radar (see Figure 6b).





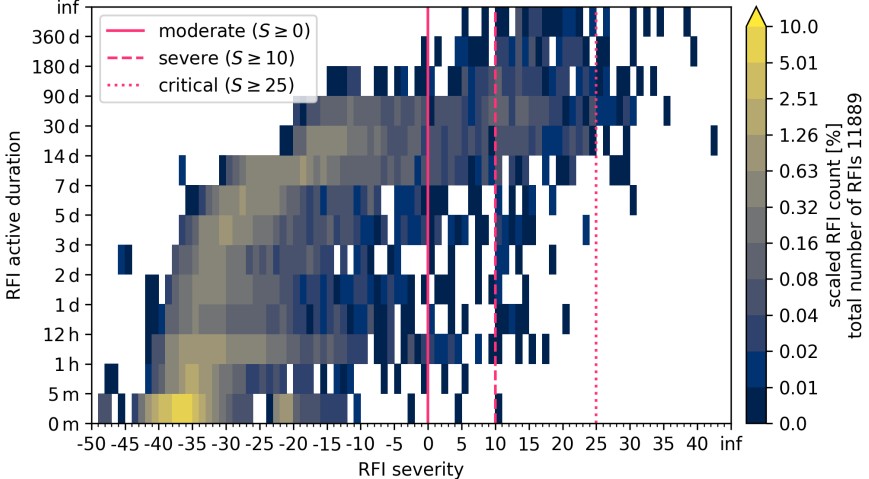

**Figure 7.** Frequency distribution of all RFIs from July 2017 to May 2022 with respect to the RFI severity and the active RFI duration. The number of RFIs in each bin is scaled by the total amount of detected RFIs (denoted on the right). Vertical lines mark RFI severity thresholds for moderate, severe and critical RFIs.

## 4   Longterm interference statistics of the radar network

This chapter contains an assessment of the interference situation for the German weather radar network since July 2017, when the systematic interference detection over the entire network was first introduced. The here introduced RFI detection algorithm is retrospectively applied over the entire available time period. DWD service availability statistics show that the radar systems were, on average, up and running for over 97 % of this period. Also, there is no significant year-to-year variability in radar availability for the analysed time period. Therefore and for simplification, the actual downtime of radar systems is ignored

in this analysis, effectively increasing the time where no interference is recorded and a clean data collection is assumed. In particular, this chapter discusses characteristics of detected RFIs in relation to their RFI severity and noticeable trends over the available time period.

  Since July 2017, in total 11889 RFI sources were detected by the presented RFI detection algorithm. Each RFI is automatically categorised according to its calculated RFI severity (see Section 2.1). In contrast to the operational RFI detection, the

severity impact factor regarding the "last activity" of an RFI is not considered in this analysis as it is only relevant for the monthly analysis under operational circumstances. The RFI severity in this analysis can therefore theoretically reach values between −55 and +80.

  The most important factor regarding a possible mitigation of a detected RFI is its active duration. Most of the detected RFIs persist for less than a day (61.58 %) and disappear without any need for an active intervention. Therefore, the relation between

RFI severity and active duration of the RFI gives the best overview of the RFI situation in relation to the mitigation process with the NRA. A heatmap can be generated to visualize this relation by selecting relevant time scales from 5 minutes to 180





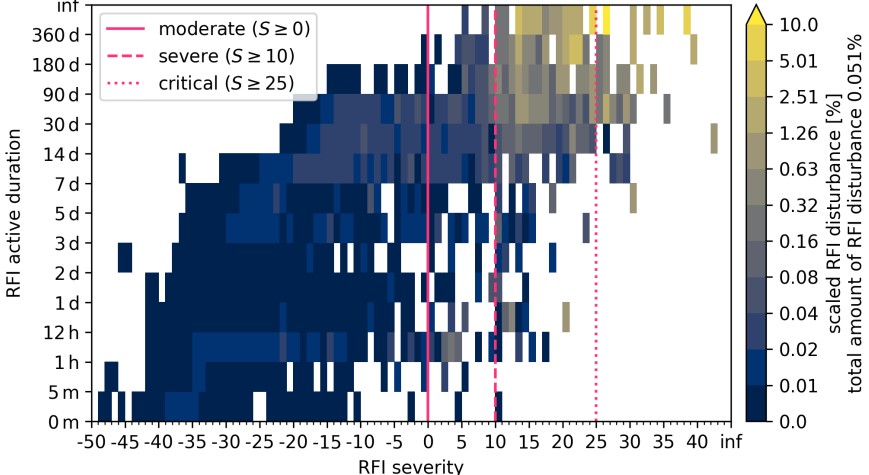

**Figure 8.** Heatmap of aggregated disturbance caused by RFIs from July 2017 to May 2022 with respect to the RFI severity and the active RFI duration. The percentage of compromised rays aggregated in each bin is scaled by the total amount of caused disturbances (denoted on the right). Vertical lines mark RFI severity thresholds for moderate, severe and critical RFIs.

days and using discrete RFI severity bins (of width 1). Figure 7 shows the number of detected RFIs in each bin scaled by the total number of detected RFIs. The vertical lines at RFI severity values of 0, 10 and 25 mark the limits for those RFIs that are treated in the mitigation process. All RFIs with a severity below zero are not considered in the active mitigation process with
380 the NRA. These untreated RFIs amount to 94.8 %, which are mostly short lived and occur sporadically in time and azimuth. Moderate RFIs amount to 1.9 %, severe ones to 2.9 % and critical ones to 0.3 % of all detected RFIs. Here, the meaning of a *detected RFI* has to be emphasised. A *single RFI* can either be short-lived (e.g. it may occur just once in a single sweep) or be active for over 360 days at a time. In particular, a persistent RFI which is continuously present for 360 days is still counted as *one RFI* in this figure.

In order to simultaneously illustrate the threat to data quality caused by these detected RFIs, the caused disturbance (daily mean fraction of disturbed rays) for each RFI together is evaluated. For the analysed time period in total 0.051 % of all (expected) rays are disturbed by an RFI signal. This means that on average 52 compromised rays occur in the course of a day in each elevation (of the analysed four) for each radar site. Figure 8 shows the scaled contribution to the caused disturbance as a heatmap following the layout of Fig. 7. Integrating over all moderate RFIs amounts to 2.5 % of the overall disturbance, severe
RFIs accumulate to 51.9 % and critical RFIs to 42.1 %. Thereby, the chosen RFI severity categorisation, with an RFI severity greater zero, captures 97 % of the overall disturbance, for which an active mitigation process is promising and intended.

RFIs with a severity below zero are the cause for only 3.4 % of the overall disturbed rays. These 3.4 % are mostly short-lived and cause only a small fraction of the overall compromised data in the sweeps analyzed for this paper. However, and this is



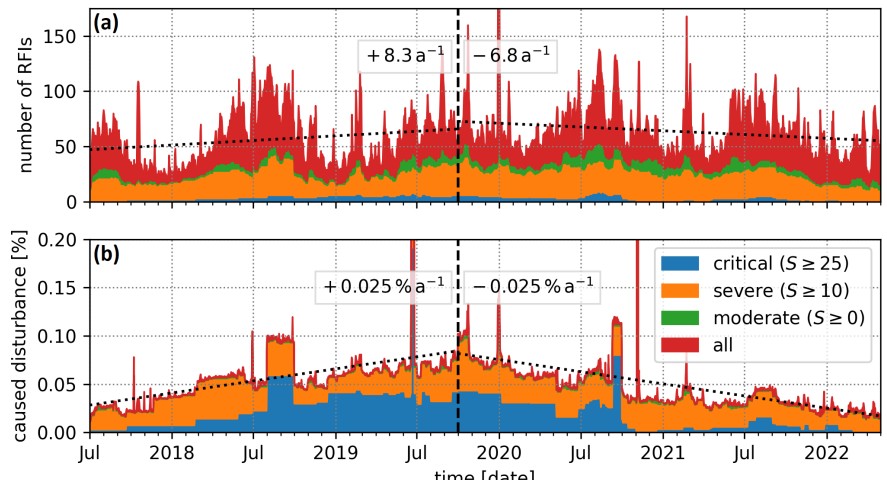

**Figure 9.** Daily timeline of detected RFI sources over the entire DWD radar network (18 sites) from July 2017 to May 2022. Each RFI active during a particular day is add to the displayed statistics for that day. Panel (a) contains the number of detected RFIs, while panel (b) contains the percentage of caused disturbance. The caused disturbance is given relative to the total amount of expected rays (288 rays in one direction during a day at each elevation for each site).

**Table 2.** Historical trends and statistics of detected RFIs by RFI category including data from July 2017 until May 2022 are given. The trends are also provided relative to the implementation date of the first automatic RFI detection in October 2019 (referenced as Oct19) The caused disturbance is given relative to the total amount of possible rays (288 rays in one direction during a day at each elevation for each site).

| RFI | | yearly trend of RFI count | | | number of RFIs | | | caused disturbance | |
|---|---|---|---|---|---|---|---|---|---|
| category | severity limit | before Oct19 | after Oct19 | overall | yearly | overall | relative [%] | absolute [%] | relative [%] |
| - | $S < 0$ | 1.7432 | 1.3142 | 1.3402 | 2331.9 | 11276 | 94.84 | 0.001742 | 3.42 |
| moderate | $0 \geq S < 10$ | 1.4462 | -1.4006 | 0.6785 | 46.7 | 226 | 1.90 | 0.001282 | 2.51 |
| severe | $10 \geq S < 25$ | 2.5794 | -4.9767 | 0.3858 | 72.4 | 350 | 2.94 | 0.026487 | 51.93 |
| critical | $25 \geq S$ | 2.5518 | -1.7446 | -0.2825 | 7.7 | 37 | 0.31 | 0.021489 | 42.13 |
| all | - | 8.3206 | -6.8077 | 2.1220 | 2458.65 | 11889 | 100.00 | 0.051000 | 100.00 |

very critical, they alone account for 94.8 % of all detected RFIs and have sometimes been active for more than 90 days. A

detailed summary of the statistics can be found on the right side of Table 2.

Another factor not captured by the heatmaps is the temporal variability and trend of the detected RFIs. Therefore, the detected RFIs are evaluated daily based on the four RFI categories since July 2017 over the entire time period (see Figure 9). The number of detected RFIs (a) and the caused disturbance (b) are shown. For both data sets, two linear regressions are





calculated including RFIs of all categories. The linear regressions are fitted from the initial start of the data collection July
2017 to October 2019, when the first version of the RFI detection algorithm was introduced, and from there until May 2022.
For each linear regression the change per year is noted in the respective section.

In both panels the effectiveness of the introduced RFI detection algorithm is clearly visible by comparing the slope of the
regression lines before and after October 2019. Also, the difference between the number of detected RFIs in each category and
the respective amount of caused disturbance is very prominent by the color coding, following the findings discussed before. In
the top panel, a seasonal variation in the number of detected RFIs is visible which is more prominent for the short-lived and
thereby sporadic RFIs below zero severity. As of now, we have not found a definitive reason for the increase in RFIs during the
summer time, but it could be caused by increased human activity outdoors or changing atmospheric beam propagation.

Even though only a few critical RFIs are active at any given day, the amount of caused disturbance is dominated by these
critical RFIs. It is particularly interesting to observe the block like structures of these critical RFIs in the lower panel of Fig. 9
before October 2019. These blocks are caused by single critical RFIs which start to disturb the radar data, are present for up
to a couple of months and finally got removed which then causes the sharp drop in disturbance. By comparing these structures
before and after the introduction of the RFI detection algorithm, one can see the most important benefit, which is the decreased
removal time of critical RFIs. In 2018, it took on average about 2 to 6 months to remove a critical RFI, while after the initial
implementation of the RFI detection algorithm in October 2019 the average time to removal decreased to a few weeks.

The decreased time to shut down an RFI source has an important implication. As the active duration of an RFI increases,
the RFI severity also increases, up to the 30 day maximum. An RFI which starts out as severe can be quickly removed and not
even become critical, leading to less caused disturbance overall. This has to be kept in mind when looking at the daily timeline
(Figure 9) or the yearly trends in Table 2. In general, a faster removal of RFIs can not decrease the total number of observed
RFIs, only the caused disturbance, which can lead to a decrease in RFI severity.

To conclude, even though we have now established a procedure to shut down persistent RFIs, their generally sporadic nature
hinders a quick mitigation and compromises data quality. The mitigation process itself is still time consuming, requiring on
average several weeks for each RFI, but has been optimized by improving the communication with the NRA in terms of
available information on each RFI and reorganizing the reporting process. Every year, the equivalent of one full-time position
at DWD has to be committed to the mitigation process to continue the successful removal of RFIs in the radar data.

Another crucial finding not clearly visible in the timeline is the steady increase in RFIs with a severity below zero. Table 2
contains a detailed analysis of the yearly trend in the number of detected RFIs for each of the RFI categories. Even after the
introduction of the RFI detection algorithm, the number of those RFIs increases by 1.3 each year, which might at first sight
seem irrelevant in respect to the 11276 RFIs in this category. However, those mostly short-lived RFIs cannot be included in the
mitigation process, as there is no chance for the NRA to identify the source of these. Also, it has to be emphasized that even
though the detection of RFIs in radar data works quite well (e.g. Cho, 2017), there are no filter methods neither in hardware
nor software available that are able to separate a superimposed external RFI signal from the meteorological signal. Depending
on the strength of the superimposed RFI signal, more or less of the meteorological information is inevitably lost.



## 5 Conclusions

C-Band weather radars are primary and therefore privileged users in their frequency band, which, however, is shared with WIFI
services. WIFI services are required to not disturb weather radars by selecting an appropriate operating frequency via DFS
(ETSI, 2012). It is the task of the national NRA to enforce the proper coexistence between weather radar and WIFI services.
If a weather radar is disturbed, the NRA has to identify the WIFI service and shut it down. To deal with the increasing number
of disturbances in the DWD weather radar network, the approach presented in this paper has been introduced. An efficient and
reliable radar data based detection algorithm for RFIs has been implemented in 2017. The automated classification of the RFIs
was added in October 2019 and was subsequently improved further.

The current RFI detection algorithm provides all the necessary information for DWD service personnel and the NRA, so that
they are able to efficiently trace persistent WIFI sources, which are present for at least a few days. If a severe RFI is detected,
which is present for about 14 days, the NRA is nearly always able to find the source of the interference and shut it down. Even
with this optimised approach, many work hours have to be put into the mitigation process, which by now has significantly
reduced the amount of disturbance caused by RFIs. The main factor contributing to the reduced disturbance is the reduction in
lifetime of severe and critical RFIs. These RFIs are now removed within days or weeks, rather than within months as before.
However, the number of appearing RFIs is still steadily increasing (only excluding critical RFIs, again partially because of
the reduced lifetime directly reducing their severity). It is also crucial to note that most severe and critical RFI sources do not
cease to exist without active intervention. Therefore, before the introduction of the automated RFI detection, they tended to
accumulate over time.

A comparison between the radar data based RFI detection algorithm and so called RHunt measurements was shown. To
achieve this, we have implemented an automatic scanning tool for the RHunt to record all identifiable WIFI networks that
operate in the 5 GHz C-Band. By comparing the recorded list of WIFI networks with the direction of detected RFIs, further
information on the most likely RFI sources is available for the mitigation process. However, an absolute validation of the RFI
detection algorithm was not possible with the utilised RHunt setup, because of the limitations introduced by the commercial
WIFI adapter.

With our almost 5 year long data set of disturbed rays, we can show that in particular the short term, sporadic RFIs are
increasing each year. The steady increase of short term RFIs is especially concerning due to the fact that they can not be
tracked, identified or controlled by DWD nor by the NRA in the field. Their sporadic nature currently creates a circumstance,
where we have to hope that they will not reappear irregularly in weekly or monthly intervals, as such RFIs are nearly impossible
to trace. But with the increasing implementation of outdoor WIFI we expect especially these hard to mitigate RFIs to occur
more often. Keeping in mind that the compromised meteorological signal can not be recovered, emphasises the need for strict
regulatory measures to enforce the expected coexistence of WIFI services and the privileged weather radar systems in the
C-Band. While current observations, especially of short-term interference, indicate a growing contradiction to the expected
peaceful coexistence. As Saltikoff et al. (2016) put it: "This [keeping WIFI devices out of the radar frequency band] must be
an ongoing effort on all levels, and we cannot rest and think "case solved.""





*Code and data availability.* Python code used for the presented analysis is available upon request from the corresponding author. To obtain polarimetric DWD radar data, please contact DWD customer relations at kundenservice@dwd.de. Basic DWD radar data can be obtained via https://opendata.dwd.de.

## Appendix A: Technical details of the RFI detection

This appendix section contains explanations about the inner workings of the RFI detection algorithm and an additional figure. First the overall process is described followed by a detailed description of how RFI boxes are joined together:

1. **preparing the data** - All disturbed rays (mean SQI below 0.6 and mean STD above 0.6) within the last 30 days are collected.

2. **combining disturbed rays** - All disturbed rays within fixed 2 hour windows are grouped, if they are no further than 5° apart in azimuth. A group of such disturbed rays is called an RFI ray box. An RFI box is defined by an upper and lower boundary in azimuth as well as a specific start and end time, resulting from the outermost disturbed rays captured in the box. Here rays are combined regardless of their elevation. Thereby, combining data from all four used scans (PCP, VOL 0.5, VOL 1.5, VOL 2.5). Due to the 2 hour window and the 360° azimuth there could be up to 12.960 RFI ray boxes within a 30 day period. As there are generally only a few specific azimuth directions affected by interference at any given time, the number of initial RFI ray boxes for all 17 DWD sites combined is usually about 6.000 for the 30 day period.

3. **joining the RFI (ray) boxes** - This step is the core of the presented RFI detection algorithm. Here, two processing steps are alternately repeated until all RFI boxes with similar properties are joined in time and azimuth. This results in a few final RFI boxes which capture the persistent and strong interference. These two processing steps are explained in detail below.

4. **characterising the RFI boxes** - All final RFI boxes are classified by an RFI class and RFI severity which are both described in Section 2.1.

5. **result visualisation and aggregation** - The final RFI boxes contain detailed information on each detected RFI source. More information on this can be found in Section 2.2. Additionally, appendix Fig. A1 shows RFI detection results over 6 months. Here one can see final RFI boxes over a longer time scale, including RFI boxes for RFIs with a severity below zero (blue in the top panel). Some blue boxes do partially overlap with the red RFI boxes without being joined, especially around 23 March at about 145° azimuth, as those RFI boxes are not similar enough to be joined (for more details see explanation below).

The process of joining all similar RFI boxes is composed of two main steps, here distinguished as "merging" (step one) and "connecting" (step two) the RFI boxes. These two steps are alternately repeated until no further changes occur. Carrying out either or both of these two steps is referred to as "joining" RFI boxes. To execute any kind of join between RFI boxes, all





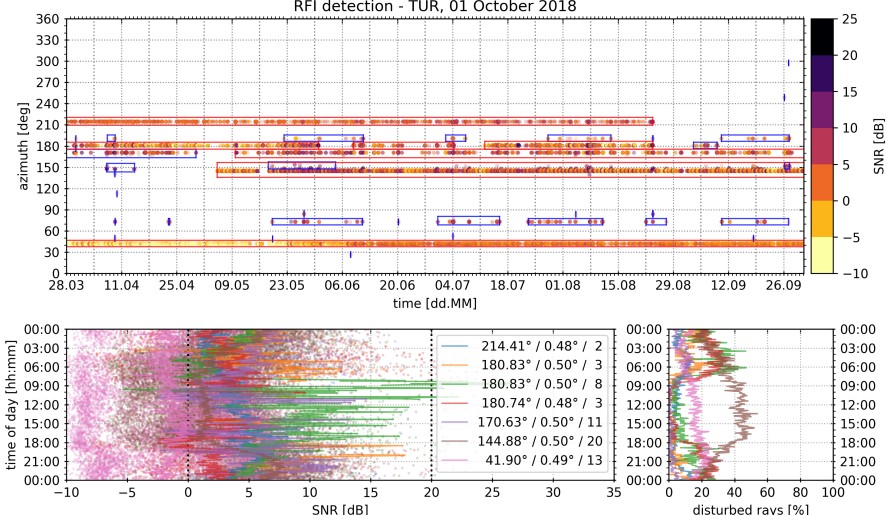

**Figure A1.** An operational RFI analysis from the radar site Türkheim (TUR) on 1 October 2018. Included are disturbed rays from 6 months. The top panel shows each disturbed ray as a dot in the scatter plot (color of dots depicts the median SNR). All detected RFIs with an RFI severity greater zero are marked with a red box, while blue boxes mark RFIs with an RFI severity below zero. At the bottom additional properties of each red RFI box are displayed. The data in both bottom panels is accumulated into 5 min bins according to the time of day (y-axis). The bottom left panel shows SNR and the bottom right the fraction of disturbed rays. Within the legend the mean azimuth, mean elevation and RFI severity of each red RFI box are displayed.

boxes, which can be joined to a specific box, are first identified and the resulting connections are evaluated before the actual join is executed. Thereby, RFI boxes which are not to be joined directly can be joined via a third box within one step of the
iteration. This is most common for the time dimension, as there is always a fixed constraint on how far in time two RFI boxes may be joined together. To execute a join of two RFI boxes, the outermost azimuth angles and times of either box are used to define the new box. Afterwards the different disturbance fractions of the resulting RFI box are recalculated. The following is a description of the algorithm used to join RFI boxes:

**similar RFI boxes** Only RFI boxes fullfilling at least one of these criteria may be joined together. If none of these three
properties are met, the two RFI boxes are most likely originating from two independent interference sources and will therefore not be joined directly.

1. The absolute difference in number of disturbed rays has to be smaller than 50. This is mainly applicable to short term RFIs and the initial RFI ray boxes.

2. The difference in mean disturbance fraction is smaller than 10 %. This is the most common similarity as the mean
disturbance fraction of most RFIs is in the order of 10 %.

3. The difference in "worst" disturbance fraction is smaller than 10 %.



**Step 1: merging RFI boxes**  Similar RFI boxes with matching azimuth boundaries are merged in a small time window.

1. The lower and upper azimuth boundaries of two RFI boxes have to line up within $\pm 2°$.

2. The two RFI boxes may only be up to 24h apart in time to be considered for merging.

**Fully enclosed RFI boxes**  There can be some RFI boxes which are not merged and which are fully enclosed after the merge step. This mostly happens if a 2 hour ray box is enclosed by two neighboring boxes that are much wider in azimuthal width (more than the allowed $2°$ on each edge). These fully enclosed boxes are dissolved into the RFI box enclosing them (without accounting for similarity). If an RFI box is fully enclosed by multiple other boxes, it is dissolved into the box with the best matching mean disturbance fraction.

**Step 2: connecting RFI boxes**  Similar RFI boxes with a similar mean direction are connected bridging long time periods.

1. The mean azimuth (weighted by SNR of each ray) of two RFI boxes has to be within $\pm 4°$.

2. The difference in azimuthal width is explicitly limited to a maximum of $10°$.

3. The two RFI boxes may only be up to 7 days apart in time to be connected.

**Fully enclosed RFI boxes**  These are again dissolved as described before.

Repeating these two steps ensures a consistent and formal way of identifying resulting RFI boxes and thereby RFI sources. In most cases, these two steps only have to be carried out once after which no further changes occur. When using this approach for long time scales (over years as in Chapter 4), some of the joined RFI boxes resulting from either of the two steps can have changed RFI box properties regarding the *similar RFI box* rule which requires further iterations of the two steps.

**Appendix B:  RHunt Data acquisition**

During the data acquisition every detected WIFI network is tagged with the respective azimuth position of the radar antenna. The following sequence of steps is carried out at $0°$ elevation to acquire the RHunt data:

1. Operational scanning is disabled.

2. The radar transmitter is set to operate with a PRF of $600\,\text{Hz}$ and a pulse length of $0.8\,\mu\text{s}$, as a transmitting radar is of course the prerequisite that the DFS of WIFI transmitters can work properly.

3. The following steps are repeated until the full sweep is acquired:

   (a) The radar antenna is moved in $1°$ steps at a constant elevation of $0°$.

   (b) Before the actual data acquisition is started, the antenna is fully stopped.





    (c) All identifiable WIFI sources are recorded with the RHunt which takes about 10 seconds. Each record is tagged with a time stamp, the azimuth position and contains all information available from the commercial WIFI receiver. This is done by using the linux command "sudo iwlist wlan0 scan" on the Raspberry Pi and gathering all of the relayed information. This generally includes the MAC-address, a signal power level, encryption method and a (possibly empty) ESSID. The exact information which can be collected depends not only on the used hardware and its capabilities, but also on the used Linux drivers. It should also be noted, that in order to get accurate real-time results, the scan must be executed with *root* permissions. Otherwise, the Linux kernel returns a cached result, which can be up to one minute old.

4. After the data acquisition of the complete sweep, the radar immediately starts with operational scanning including the respective RFI detection. Thereby, enabling a direct comparison between the RHunt data and the raw radar data.

*Author contributions.* MS implemented the RFI detection algorithm, the subsequent optimisations, the RHunt data processing and wrote this paper. MF setup the interference monitoring in 2017, supported the entire project, edited the text and mentored especially during the early drafting phase. DM implemented the hardware control for the RHunt measurements and added details regarding the RHunt data acquisition. CH contributed the basis for the PPI visualisations and helped edit the text. BR edited the text, added details regarding WIFI regulations and the RHunt setup.

*Competing interests.* The DWD is a member of OPERA (the radar program of EUMETNET). In OPERA a European interference detection is currently setup by Vera Meyer (ZAMG). The authors declare to have no further competing interests.

*Acknowledgements.* I am very grateful for the valuable input from my colleagues at DWD whose contributions made this paper much more concise. The support of this group of people allows for a very motivating, goal oriented and productive workflow. The presented RFI detection algorithm is based on the approach utilised in the GAMIC signal processor to use the STD moment and respective thresholds to reliably detect interference in radar data. The RHunt hardware setup was inspired by AustroControl, who use a similar setup in their radars. Contributions from Maximilian Schaper were supported by the DWD Innovation in Applied Research and Development (IAFE) program under project 'VH 3.10 Radarmonitoring'.



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
