# Peer review of "RF-interference detection and mitigation in the DWD C-Band weather radar network"

_EGUsphere, 2022_

## Author Response (AR1)

**Referee comment 1 and response:**

**Regarding the general comment:**

This paper describes an automated objective procedure for monitoring and prioritizing RFI in German C-band weather radar data. Results are provided, which show that the method works effectively, and that it helps the national regulation agency to track and shut down the offending sources. The procedure presented here should be of considerable practical interest to operators of C-band weather radars in other countries where RFI is a problematic issue. I recommend the publication of this paper after some minor issues are addressed and improvements are made to the English language (grammar, punctuation, etc.)

> All requested issues are addressed below and improvements to the English language have been made.

**Regarding the specific comments:**

Line 78: Does the SQI computation really involve correlation of received signal against the transmitted signal? My understanding of SQI is that only the received signal is processed (magnitude of autocorrelation lag 1 divided by autocorrelation lag 0).

> Line 78: It is correct, that the computation of SQI does not involve the correlation of the Rx-signal with the Tx-signal, but the so called "coherence" with the Tx-pulse is intrinsic to the retrieved moment. When calculating the radar moments from IQ-data the Tx-phase is crucial. Therefore, the random phase of each Tx-pulse is recorded (random each pulse due to magnetron drift). During processing, the measured IQ-data is shifted in phase to cohere to the Tx-pulse, which has a major impact on the resulting moments, as it effects the separation of signal and noise. This step is necessary as the auto correlation lag one is not calculated by shifting a single pulse (IQ time series) in range, but by correlating successive pulses in the range gate. Therefore, a reasonable correlation can only be retrieved, if both Rx-signals are correctly aligned based on their individual Tx-phase. To achieve this, each Rx-pulse has to be coherent to its Tx-pulse. In particular, this effect is commonly used to identify and eliminate 2nd and multi trip echoes. If the utilized Tx-phase does not match the Rx-signal, the signal appears as white noise in the power spectrum due to its random phase. A meaningful signal is only retrieved if the measured IQ-data is shifted correctly to cohere to the Tx-pulse. In other words, a non-coherent Rx-pulse will always lower the retrieved SQI, even though the calculation is done by solely correlating successive pulses. As such, any external interference signal will be non-coherent to the transmitted radar pulse and thereby, it will increase the spectral width and reduce the SQI. On the other hand, a coherent external signal would be interpreted as valid weather signal with a high SQI. - Clarified marked sentence: "SQI quantifies the coherence..." to "SQI effectively quantifies the coherence...".

Line 81: "…50 pulses are aggregated within one ray" Add "at the same range gate"

> Line 81: added reference to range gates, now: "...about 50 pulses are aggregated within one ray for each range gate..."

Line 93: What is "uncorrected reflectivity" (as opposed to corrected reflectivity)?

> Line 93: "uncorrected reflectivity" highlights the fact that there are no filters applied at all, added: "In (a) uncorrected reflectively (UZ) is shown, which is derived from the received power without applying a clutter filter or a thresholding."

Line 100: How did you decide on the thresholds? It would be more objective if you could state that XX% of weather signals are above the SQI threshold or below the STD threshold, while YY% of RFI signals are below the SQI threshold and above the STD threshold.

> Line 100: The SQI and STD thresholds are taken from the definition used in the GAMIC signal processor ENIGMA. There they are used to identify and flag external interference. No further effort was taken to verify the chosen values as the detection seems reasonable (see RHunt verification). Therefore I am not able to provide any further statistical insight. However, it is important to note that, as of now, one can not find the "YY\%" as requested in the question "...while YY\% of RFI signals are below the SQI threshold and above the STD threshold.", because as of now there are no continuously available, independent data to verify if an RFI signal was/is present. Only the chosen thresholds of SQI and STD yield the classification of RFI contaminated rays based on the radar data itself.

Figure 3: What is the significance of the dashed lines at SNR = 0 and 20 dB?

> Figure 3: The dashed lines at SNR of 0 dB and 20dB mark the limits for "persistent" and "strong" RFI sources regarding the RFI class (see also table 1). Added: "Dashed lines in the left panel correspond to RFI class SNR limits (persistent and strong)."

Figure 4: What is TSG?

> Figure 4: TSG references the built-in "test signal generator" used to calibrate the receive path of the radar. Added: "Also shown is the test signal generator (TSG), which is used to calibrate each receive path."

Figure 8: The unit here is confusing. The scale is in %. Percent of what? The total amount is given as 0.051%, so how is the plotted values scaled (i.e., what is the denominator)?

> Figure 8: The amount of disturbance caused by RFI sources (percentage of compromised rays) is accumulated in each bin. The result is than scaled by the total percentage of caused disturbance. The plotted unit is therefore a scaled percentage (percent scaled by total percent).

Line 385: Simultaneous between what and what? Delete "simultaneous" or state what those things are.

> Line 385: rewrote the sentence for clarification: "In order to relate the pure number of detected RFI sources to an actual threat to data quality caused by them, the disturbance caused by each RFI source is evaluated as a daily mean fraction of disturbed rays."

Figure 9: What is "$a^{-1}$"?

> Figure 9: "a\^-1" means each year and the value is thereby the inclination of the fitted line.

Line 479: Shouldn't the number of ray boxes be an integer?

> Line 479: This is a spelling issue, now: "12,960 RFI ray boxes" instead of "12.960 RFI ray boxes"

Line 481: Get rid of decimals.

> Line 481: This is a spelling issue, now: "about 6,000" instead of "about 6.000"

**RC1 technical Comments:**

- There is a mix of American (summarize, recognize, normalize, visualize, minimize, standardize, utilize, optimize, reorganize, emphasize, program, etc.) and British (prioritise, utilise, categorise, localise, etc.) spelling. Pick one and stay consistent. British spelling is recommended, since this is an European journal.
- Use the term "Section" instead of "Chapter" throughout. This is an article, not a book.
- Replace all instances of "dynamic frequency stepping" by "dynamic frequency selection."
- "RFI" should not be pluralized, so replace "RFIs" with "RFI cases" throughout.
- Replace "WIFI" with "Wi-Fi" throughout.
- Replace "short lived" with "short-lived" throughout.
Line 7: "interference" --> "radio frequency interference"
Line 8: "…network is operational since July 2017, which makes…" --> "...network, operational since July 2017, makes..."
Line 9: Delete "the"
Line 13: "(BNetzA)" --> "(Bundesnetzagentur, BNetzA)"
Line 15: 94.8% is more than just a majority. Change "The majority" to "Most"
Line 31: Delete "especially"
Line 35: No need to spell out what RFI means again here.
Line 39: BNetzA now spelled out earlier, so need to do so again here.
Line 39: "starts to act upon" --> "acts upon"
Line 50: "exists" --> "exist"
Line 150: "there" --> "their"
Line 185-186: "above zero" --> "greater than or equal to zero and less than ten"
Line 188: "greater than 10" --> "greater than or equal to 10 and less than 25"
Line 189: "greater 25" --> "greater than or equal to 25"
Line 196: No need to spell out ISN here and in subsequent instances.
Line 226: "has to antenna ports" --> "has two antenna ports"?
Line 246: DFS already spelled out previously, so no need to do so again here.
Line 251: "radar free" --> "radar-free"
Line 253: "arises" --> "arises, as in the following situations"
Line 255: Suggest changing to "The geolocation is deliberately or unintentionally set incorrectly, or the automatic setting is incorrect."
Line 267: "Thereby, including many…radar." --> "Thereby, many...radar are also included."
Line 280: No need to spell out MHP again here.
Line 281: "next bigger"? Do you mean the biggest nearby cities?
Line 281: "in" --> "to"
Line 282: "from the radar" --> "in azimuth from the radar"
Line 287: "outdoor" --> "outdoors"
Line 314: Delete the opening "The"
Line 328: "to" --> "too"
Line 336: Delete "used"
Line 338: Delete "i.e."
Line 340: "transmitter" --> "transmitters"

Line 347: "allows to detect any" --> "allows the detection of any"
Lines 355-356: "allow to draw a definitive connection" --> "allow a definitive connection to be drawn"
Line 356: "caused interference" --> "and the resulting interference"
Line 356: Delete "which would relate them to the detected RFI"
Figure 7: moderate: $0 <= S < 10$, severe: $10 <= S < 25$
Line 359: "Longterm" --> "Long-term"
Line 361: "here introduced RFI detection algorithm" --> "RFI detection algorithm introduced here"
Line 373: 61.58% is not most. Replace "Most" with "A majority"
Line 391: What is meant by "promising" here? Perhaps replace by "promised" or delete?
Figure 9: severe: $10 <= S < 25$, moderate: $0 <= S < 10$
Line 408: "at" --> "on"
Line 409: "block like" --> "block-like"
Line 411: "got" --> "get"
Line 422: "average several" --> "average of several"
Line 422: "improving the" --> "improving"
Line 428: "in respect" --> "with respect"
Line 447: Delete "only"
Line 458: "short term" --> "short-term"
Figure A1: "greater zero" --> "greater than or equal to zero"
Line 510: "in" --> "on"
Line 530: "During the data" --> "During data"
Line 540: "linux" --> "Linux"

**>>> Regarding the technical comments:**

Changed American "ze" spelling to "se" everywhere.
Changed "chapter" to "section" everywhere.
Changed all mentions of DFS to dynamic frequency selection.
Replaced "RFIs" with "RFI sources" everywhere.
Replaced "WIFI" with "Wi-Fi" everywhere.
Changed spelling of "short-lived" and "long-lived" everywhere.
Line 7-189: Added all suggestions.
Changed redundant mentions of "Isen (ISN)" to "ISN".
Line 226: Changed as suggested.
Line 246: I would prefer to keep it spelled out here, as this is where the DFS is explained in detail.
Line 251-267: Added all suggestions.
Changed redundant mentions of "Hohenpeißenberg (MHP)" to "MHP".
Line 281-356: Added all suggestions.
Figure 7: Changed legends of Figure 7 and Figure 8 as suggested.
Line 359-373: Added all suggestions.
Line 391: rewrote as suggested, now: "...for which an active mitigation process is expected to identify the RFI source."
Figure 9: Here the values shown are accumulated - changed legend to: ">= moderate (S >= 0)", ...
Line 408-411: Added all suggestions.
Line 422: Sentence kept as is: here "on average several weeks" is meant not "an average of several weeks"
Line 422-458: Added all suggestions.

Figure A1: Changed as suggested.
Line 510: Sentence kept as is: here a distance in the time dimension is meant
Line 530-540: Added all suggestions.

**Referee comment 2 and response:**

**Regarding the general comment:**

A method to objectively identify and classify RFI is described in the paper for German c-band radar network. The method work proprely and using it in close collaboration with Gernam NRA leave to promising results.

I recommend the pubblication of the paper after minor revision.

As a general comment the description of section 2.1 is not so clear, so a more organised anc cleaned presentation of how an RFI is identified anc classified is needed.

> The description of the RFI identification and classification is summarized in the enumeration at the beginning of section 2.1. A detailed description of the exact method used to identify an RFI source is given in appendix section A. The classification is done in two steps. First an RFI class is determined for each identified RFI source. Afterwards the internal parameters derived during the identification are used, together with the RFI class, to determine the RFI severity of each identified RFI. To clarify the classification process multiple examples are given throughout section 2.1 regarding the RFI class and RFI severity.

**Regarding the minor comments:**

1) line 22 - The sentence "Forecast models assimilate radar data." need to be better integrated in the text.

> Line 22: rewrote the sentence, now: "Many forecast models assimilate radar data, successfully improving their prediction skills."

2) line 25 - (Deutscher Wetterdienst, DWD: - I don't understand why insert this. explain plase

> Line 25: "German Weather Service" is the English term. The equivalent long German term is "Deutscher Wetterdienst", which is why the abbreviation "DWD" is commonly used in Germany to reference the national weather service.

3) line 29 - add **hereinafter** before the acronym WRC-03

> Line 29: Removed the abbreviation entirely as it is not used anywhere.

4) line 35 - add **hereinafter** before the acronym RFI

> Line 35: Removed the abbreviation as of the recommendation of the first referee comment.

5) figure 4 what means TSG?

> Figure 4: TSG references the built-in "test signal generator" used to calibrate the receive path of the radar. Added: "Also shown is the test signal generator (TSG), which is used to calibrate each receive path."

6) figure 8 Is not so clear what represent the color scale on the left? wht is the units?

> Figure 8: The amount of disturbance caused by RFI sources (percentage of compromised rays) is accumulated in each bin. The result is than scaled by the total percentage of caused disturbance. The plotted unit is therefore a scaled percentage (percent scaled by total percent).

Line 385: rewrote the sentence for clarification: "In order to relate the pure number of detected RFI sources to an actual threat to data quality caused by them, the disturbance caused by each RFI source is evaluated as a daily mean fraction of disturbed rays."